# Designing Environmental Messages to Discourage Red Meat Consumption: An Online Experiment

**DOI:** 10.3390/ijerph19052919

**Published:** 2022-03-02

**Authors:** Alice Wistar, Marissa G. Hall, Maxime Bercholz, Lindsey Smith Taillie

**Affiliations:** 1Program in Global Health and Health Policy, Center for Health and Wellbeing, Princeton School of Public and International Affairs, Princeton University, Princeton, NJ 08544, USA; alicewistar@gmail.com; 2Department of Health Behavior, Gillings School of Global Public Health, University of North Carolina at Chapel Hill, Chapel Hill, NC 27599, USA; mghall@unc.edu; 3Lineberger Comprehensive Cancer Center, University of North Carolina at Chapel Hill, Chapel Hill, NC 27599, USA; 4Carolina Population Center, University of North Carolina at Chapel Hill, Chapel Hill, NC 27516, USA; bercholz@email.unc.edu; 5Department of Nutrition, Gillings School of Global Public Health, University of North Carolina at Chapel Hill, Chapel Hill, NC 27599, USA

**Keywords:** vegetarianism, goal framing, emphasis framing, sustainability, meat consumption, environmental behavior, health communication

## Abstract

Reducing red meat consumption in high-consuming countries is critical for mitigating climate change and preventing chronic disease. This study tested the effectiveness of messages conveying the worsening or reduction of environmental harms at discouraging red meat consumption. 1078 U.S. adults viewed seven messages in an online survey highlighting the reduction or worsening of environmental harms associated with eating red meat (between-subjects factor) and rated the messages on how much they discouraged them from wanting to buy beef. Each message highlighted a different environmental harm: deforestation, climate change, water shortages, biodiversity loss, carbon footprint, greenhouse gas emissions, or environment (within-subjects factor). No statistically significant difference was found between the reduction and worsening of environmental harms conditions for most topics, though the worsening of harms frame slightly outperformed the reduction of harms frame for the ‘environment’ topic. ‘Environment’ was also the message topic that elicited the strongest response from participants overall. Latino participants, those with more than a high school degree, and those who consume beef once a week or less rated messages as more effective than non-Latino participants, those who completed high school or less, and those who consumed beef more than once a week. Future research should explore the effect of messages on behavioral outcomes.

## 1. Introduction

Reducing red meat consumption in countries with high levels of consumption was identified by the 2019 EAT-Lancet commission as a strategy that would benefit both the environment and human health [1]. Recent meta-analyses show the associations between red meat (beef, lamb, pork, and other mammalian meat) and colorectal cancer [2,3,4], type II diabetes [5], stroke [6], coronary heart disease [6], heart failure [6], obesity [7], and all-cause mortality [8,9]. Because of these relationships, organizations including the American Cancer Society recommend limiting red meat consumption in favor of poultry, fish, or plant-based proteins [10,11]. Substituting whole, plant-based foods such as nuts, legumes, and whole grains for red meat is associated with a lower risk of type II diabetes [12,13] and mortality risk [14] and can lead to an increase in life expectancy [15]. One meta-analysis concluded that reducing or eliminating consumption of red or processed meat would also reduce the risk of stroke, coronary heart disease, and heart failure [6]. 

In addition, livestock contributes 14.5% of all anthropogenic GHG emissions [16], beef alone representing 41% of these emissions [16], and is the single most significant source of methane emissions [17]. Livestock also uses 30% of the planet’s ice-free terrestrial surface [18] and is a leading cause of biodiversity loss [16,17,18,19], deforestation [16,17,18,20,21], aquifer depletion and water shortages [17,22], eutrophication [23], terrestrial acidification [23], and land degradation and pollution [16,17,19,24]. 

In the U.S., where beef consumption is the second highest in the world [25] and where beef accounts for nearly half of land-use and GHG emissions associated with diets [26,27], reducing red meat consumption could yield significant environmental benefits [19,26,27,28,29,30,31,32,33,34]. An analysis from the World Resources Institute showed that reducing beef consumption to the world average level in regions such as the U.S. where beef consumption is above average could spare 300 million hectares of pasture [27], an area nearly the size of India. More recent research shows that annual agricultural production emissions of high-income countries’ diets could be reduced by 61% if their populations adopted the EAT-Lancet planetary health diet [35], which limits red meat consumption to 98 g per week [36]. A gradual phase out of all of animal agriculture, while unlikely, could achieve half of the net GHG emission reductions necessary to limit global warming to 2 °C above preindustrial levels, a goal set at the Paris Agreement, with the phasing out of beef accounting for 47% of those GHG reductions [37]. Indeed, limiting global temperature rises to 2 °C above preindustrial levels is not possible without GHG reductions from the food system [38]. Other specific environmental benefits of diets lower in red meat include reduced clearing of agricultural land [39], global water consumption [22], extinction risk for mammals and birds [39], terrestrial acidification [23], and eutrophication [23]. 

Given high levels of red meat consumption in the U.S., one challenge relates to identifying effective strategies to reduce consumption. Messaging has been an effective strategy to change health behaviors such as smoking [40], sugar-sweetened beverage (SSB) consumption [41], and alcohol use [42]. Although most public health messages about food have focused on individual health outcomes, messages that focus on the environment are promising. Considering that public awareness and concern around the threats of climate change are increasing [43], particularly among young people [44], environmental messaging may further increase awareness about red meat’s environmental harms. In an online experiment with 590 German adults, Cordts et al. (2014) found that providing information about meat’s environmental harms increased participants’ intention to reduce their meat consumption in the future [45]. Thus, environmental messaging may also help change meat consumption behaviors. Yet, relatively little is understood about the types of messages that are most effective at discouraging red meat consumption. 

Message framing, or the particular way in which information is communicated and emphasized, is an important tool used to maximize the impact of messages on individuals’ thoughts and behaviors. Though little research explores framing in the context of meat reduction messaging strategies specifically, message framing has been shown to alter individuals’ behavioral intentions and attitudes on environmental issues more broadly [46,47]. 

There are numerous types of message framing. Goal framing is one such type that involves emphasizing either the advantages of engaging in a target behavior (e.g., composting can help the environment) or the disadvantages of not engaging in a target behavior (e.g., not composting can hurt the environment) [48]. Prior research suggests that negatively framed messages are more effective than positively framed messages due to a negativity bias wherein individuals tend to dislike losses more than equivalent gains [49,50,51]. A recent systematic review of 61 studies finds that emphasizing the negative outcomes of an environmentally relevant decision is more likely to prompt a change in behavior and intentions to change behavior, whereas emphasizing the positive outcomes of an environmentally relevant decision is more likely to change attitudes towards the specified behavior [52]. Indeed, additional research conducted in the U.K., Iran, and Australia, found that messages highlighting the positive consequences of engaging in climate change mitigation activities were more effective at changing attitudes towards participating in pro-environmental behaviors than messages emphasizing the negative consequences of not engaging in climate change mitigation activities [53,54,55]. Thus, for promoting actual engagement in pro-environmental behaviors, existing research suggests that highlighting negative consequences tends to be more effective, although there is still no clear consensus. To our knowledge, no research has explored whether emphasizing positive or negative outcomes is more effective at discouraging red meat consumption specifically.

A separate type of message framing, emphasis framing, is focused on emphasizing a specific aspect or feature of a given issue (e.g., portraying climate change as an environmental issue versus a public health issue) [46]. A recent meta-analysis found that framing climate change as an opportunity for economic growth, an environmental hazard, or a moral issue modestly altered individuals’ support for climate policy and intentions to engage in environmental behaviors [46]. It is currently unknown, however, whether emphasizing specific topics within the environmental frame at large (e.g., water shortages, biodiversity loss) affects individuals’ intentions to participate in pro-climate behaviors such as reducing red meat consumption. 

In addition, little is currently known about which audiences will be most responsive to meat-reduction focused messages. While existing messaging studies in high-income countries show that being female [56,57,58,59,60,61] and eating meat less regularly [56,57,58] tend to be consistent predictors of willingness to eat less meat, there is limited and inconsistent evidence regarding the effects of other demographic characteristics such as age, education, ethnicity, and income level [62]. 

The primary objective of this research is to explore the efficacy of goal framing for discouraging red meat purchases by experimentally testing whether messages that convey the worsening of environmental harms or the reduction of environmental harms are perceived as more effective at discouraging beef purchases. Perceived message effectiveness is a predictor of longer-term behavior change [63] and is a widely-used measure in initial testing of messages because it tends to be sensitive to small differences between similar messages [64,65,66,67,68]. Given the paucity of research on goal framing for messages focused on discouraging red meat consumption, this was an exploratory study and as such, we did not have an a priori hypothesis about which type of message would elicit higher effectiveness. As secondary goals, this study explores whether PME varies between seven different environmental topics (emphasis frames) or based on demographic characteristics. 

## 2. Materials and Methods

### 2.1. Sample 

Survey participants (*n* = 1088) were recruited through convenience sampling in October 2019 using CloudResearch Prime Panels as part of an experiment that tested the impact of sugar-sweetened beverage (SSB) warnings on SSB purchasing behavior and reactions among Latino and non-Latino parents. Prime Panels used purposive sampling to recruit half Latino and half non-Latino participants.

Utilizing online panels to recruit convenience samples has been found to generate experimental findings comparable to those from representative samples [69,70]. Inclusion criteria included residing in the U.S., being 18 or older, and having at least one child between 2 and 12. The latter criterion was included because the primary experiment focused on reactions to sugar-sweetened beverage warnings among parents. Participants received incentives in cash, gift cards, or rewards points from Prime Panels. The University of North Carolina (UNC) Institutional Review Board approved this study (IRB #19-0277).

### 2.2. Messages

Message topics were chosen based on a high level of peer-reviewed and scientific organizations’ (including the Intergovernmental Panel on Climate Change) evidence on the environmental harms of beef consumption and production. For each of the seven topics, message text was then developed to either highlight that purchasing beef can *worsen* environmental harms (“worsening harms” goal frame) or that purchasing less beef can *reduce* environmental harms (“reducing harms” goal frame). Message topics (the emphasis frames) were the within-subjects factor (Table 1). Participants were randomized to see messages in either the worsening harms or reducing harms frame (between-subjects factor). 

### 2.3. Procedures

This experiment was conducted online using Qualtrics and employed a mixed between-within design. After completing informed consent, eligible participants first answered a series of questions about SSB warnings and SSB consumption. Participants then responded to one question about their beef consumption frequency before viewing seven randomly ordered messages about different environmental harms. For each of these seven messages, participants answered the same question about how much the message discouraged them from wanting to buy beef. Finally, participants answered several demographic questions. 

### 2.4. Measures 

Beef consumption frequency was measured using a modified NHANES item: “Thinking about the last month, how often did you typically consume beef (including steak, ground beef, or other types of beef)?” Response options included Never, Once a month, 2–3 times a month, Once a week, 2–6 times a week, Once a day, or More than once a day [71]. Discouragement from buying beef was measured using one item from the UNC PME Scale [72]: “How much does each statement discourage you from wanting to buy beef?” Response options were: not at all (coded as 1); a little bit (2); somewhat (3); quite a bit (4); and a great deal (5). PME is a common measure used in health-related message development and testing and has been found to predict behavioral change [73]. 

### 2.5. Analysis

Statistical analyses were conducted in Stata/SE 16.1 using two-tailed tests and a significance level of 0.05 (StataCorp LLC, College Station, TX, USA). A multilevel mixed-effects linear model with random intercepts at the respondent level was estimated to test for differences in PME between goal framing conditions (worsening of environmental harms vs. reduction of environmental harms) and message topics (climate change, deforestation, etc.). The dependent variable was PME and the independent variables were dummy variables for the worsening of harms framing condition (relative to reduction of harms) and for six of the seven message topics (relative to carbon footprint), and interactions between the worsening of harms framing condition dummy and each of the six message topic dummies. Using the model estimates, we then estimated the mean PME by framing condition on average over the seven message topics, and the difference in these means (the average framing effect). Similarly, we estimated the mean PME by message topic on average over the two framing conditions, and all pairwise differences in those means. To test whether the framing effect differed by message topic, we carried out a Wald test of joint significance of the coefficients on the interaction terms, with joint significance indicating that the framing effect did differ by message topic, i.e., was not homogenous across message topics. In further analyses, we estimated the average framing effect for each message topic separately. Results using a multilevel mixed-effects ordered logit model did not differ in any meaningful way with respect to the direction of findings and statistical significance.

To explore predictors of PME, PME was averaged over all seven topics and regressed using ordinary least squares (OLS) on categories of age, beef consumption frequency, education level, income level, Latino ethnicity, and gender. Demographic characteristics were categorized as follows—age: 18–29, 30–39, or over 39 years old; beef consumption frequency: eats beef more than once a week or less than once a week; education level: more than high school or less than high school; Latino ethnicity: yes or no; gender: man, woman, or transgender; and income level: less than $35,000, between $35,000 and $74,999, or $75,000 and above. These categorizations were chosen to create the most equal split between categories. Where an equal split was not possible in two categories (e.g., age and income level), three categories were created and two regression coefficients were reported.

## 3. Results

### 3.1. Demographic Characteristics 

Participants’ mean age was 35.3 years (Table 2). More than half (58.3%) identified as female and approximately half (47.7%) were Latino, had a college degree or higher (52.5%), and had an annual household income of less than $50,000 (46.5%). Regarding beef consumption, most (65.0%) participants ate beef once a week or more. 

### 3.2. Message Framing Conditions

The messages for both the worsening harms and reducing harms frames are shown in Table 1. The mean PME was 2.80 (95% CI 2.70, 2.91) in the reducing harms framing condition and 2.89 (95% CI 2.78, 2.99) in the worsening harms framing condition. Although this difference was not statistically significant (*p* = 0.277), it is notable that PME was on average higher in the worsening of harms frame than in the reduction of harms frame for all message topics (Figure 1). Figure 1 shows the mean PME for both messaging frames across each environmental message topic.

The coefficients on the interactions between the worsening of harms frame dummy and the message topic dummies were jointly statistically significant (*p* = 0.004), rejecting the hypothesis of a homogenous framing effect across message topics. This result was driven by the mean PME difference between frames on ‘environment’ messages: 3.01 for the message ‘buying beef can hurt the environment’ vs. 2.81 for the message ‘buying less beef can help the environment’ (*p* = 0.017), while it was not statistically different from zero for the other message topics.

Mean PME ratings by message topic are displayed in Figure 2, and ranged from 2.81 (climate change, water shortages) to 2.91 (environment). The ‘environment’ message elicited statistically higher PME (*p* < 0.05) than all topics except GHG emissions. All other comparisons between topics were not statistically significant. Figure 2 shows the mean PME, averaged across both message frames, for each environmental message topic.

### 3.3. Demographic Predictors of PME

The results of the OLS regression of average PME (averaged over the seven questions) on demographic characteristics are presented in Table 3. 

Messages elicited lower PME among participants aged 40 years and older compared to participants aged 18–29 (*b* = −0.10, *p* < 0.05). Messages also elicited lower PME among participants who consumed beef more than once a week compared to those who consumed beef once a week or less (*b* = −0.10, *p* < 0.01). Participants who had completed more than a high school degree had higher PME ratings than participants who had completed high school or less (*b* = 0.10, *p* < 0.01). Messages also elicited higher PME among Latino participants compared to non-Latino participants (*b* = 0.09, *p* < 0.01).

## 4. Discussion

This study adds to the limited literature on environmental messages aimed at reducing beef consumption. This study found no statistically significant differences in PME between messages that highlighted the worsening or reduction of environmental harms. However, the pattern of results suggested that PME tended to be higher for messages that focused on worsening environmental harms compared to reducing environmental harms, although these differences were small. The exception was the message that emphasized that purchasing beef can “hurt the environment,” which elicited greater PME than the message emphasizing that purchasing less beef can “help the environment.” The message topic that elicited the strongest response from participants was the one that included the ‘environment’ topic and messages overall elicited the highest PMEs among participants that were Latino, less than 30 years old, ate beef once a week or less, and had a college or advanced degree.

The finding that the message frames describing either the worsening or reduction of environmental harms were equally discouraging is in contrast to existing research that has explored the impacts of goal framing in environmental messages. In a systematic review of 61 studies that investigated the use of goal framing to promote pro-environmental behaviors and attitudes, only 6 found that the effect of emphasizing negative or positive outcomes did not differ, whereas 30 studies found that emphasizing negative outcomes was more effective and 18 studies found that emphasizing positive outcomes was more effective [52]. Numerous other studies have also found that either highlighting positive consequences [53,54,55,74] or negative consequences [75,76] of a decision is more effective at promoting participation in environmentally friendly behaviors. Differences in the message content could help explain the contrast in findings, this being the first study to explore goal framing to discourage beef purchases specifically. Overall, the result that the two message frames equally discouraged beef purchasing across most topics suggests that multiple framing strategies may be effective for campaigns and other initiatives aimed at generating awareness about the environmental harms of red meat.

The message that included the ‘environment’ topic elicited greater discouragement than all other topic but ‘greenhouse gas emissions’ although the difference was small in magnitude. These findings suggest that employing a more general, straightforward term like ‘the environment’ is more persuasive for people who are less familiar with specific environmental harms like deforestation and water shortages. Further, the ‘environment’ is an overarching term that encompasses the specific harms. Thus, it may seem more important or salient to consumers compared to more specific harms.

The interaction of framing (reducing harms and worsening harms) with topic was driven by differences in framing with the ‘environment’ message topic. This topic had higher PME ratings when the message focused on the worsening of environmental harms as opposed to the reduction of environmental harms. This result is consistent with environmental communications research which has found that highlighting negative outcomes is more effective at promoting participation in pro-climate behaviors than highlighting positive outcomes [52] and supports the notion of a negativity bias wherein individuals dislike losses more than equivalent gains [49,50,51]. This finding is also consistent with SSB research, which has generally found that messages emphasizing the negative consequences (e.g., graphic warning labels) of SSB consumption are more effective than health-promoting logos at getting consumers to select lower sugar products [77,78,79]. We did not see this pattern among any other message topic besides the environment. Given that the awareness of environmental harms of red meat is low [62,80], the lack of variation in PME ratings across framing conditions for the other environmental topics could be due to participants having less baseline knowledge about those specific environmental harms, whereas helping or hurting the environment was easier to understand and interpret across frames. Future research should further explore the use of the term “environment” in messages aimed at reducing red meat purchasing and consumption.

This experiment is the first that has tested the differences between Latino adults and non-Latino adults in messages related to reducing beef purchasing. Considering U.S. Latino populations consume the highest level of unprocessed red meat [81] compared to other ethnic groups, this demographic is important to assess. The finding that Latino adults rated messages as more discouraging compared to non-Latino adults is consistent with other research about Latino adults and the environment in general: Latino adults in the U.S. are more likely than non-Latino white adults to believe climate change is a problem that is human caused [82,83] and personally affecting them [82,83,84,85,86]. Considering a recent report found that individuals that were convinced global warming is happening, human caused, and an urgent threat were more willing to adopt a more plant-based diet [80], it is possible that Latino participants in our sample were more discouraged from buying red meat because of their stronger pro-environmental views and belief in human-caused climate change. Qualitative research could help further elucidate these relationships.

Our other findings examining the association between sociodemographic characteristics and PME ratings of the messages were consistent with previous literature. For example, the finding that participants below 30 years of age had higher PME ratings on average is consistent with research that has found that young people are more likely to know about the environmental harms of meat [87] and to eat less meat for environmental reasons [88]. People aged between 41 and 60, in contrast, are more likely to reduce meat consumption because of health concerns [88], suggesting that they may be less persuaded by environmental arguments to purchase less meat.

The result that participants that eat beef once a week or less were more discouraged by the messages compared to more regular beef consumers is consistent with prior research that has found that more frequent meat-eaters are less willing to reduce their meat consumption [56,57,58]. This finding also builds on research showing that reactance or resistance to persuasive messages may be more pronounced among those with higher engagement with the target behavior [89]. This difference could be due to more regular meat-eaters’ greater preference for meat [90], enjoyment of meat [91], or belief that eating meat is a natural and normal human behavior [91]. This finding also supports the notion of a ‘meat paradox,’ wherein more regular meat-eaters resist information about the harms of meat consumption because they are in conflict to their values or behavior [91,92]. Regular beef-eaters in our sample may have minimized the potential benefits of purchasing less beef instead of accepting personal responsibility, lowering PME. Belief that consuming meat is necessary for maintaining good health has also been identified as a persistent barrier for meat reduction among individuals from the U.S. and U.K. [91,93,94] despite research showing that balanced vegetarian diets are nutritionally adequate and healthful [95] and research documenting the significant health benefits that can result from adhering to diets higher in plant-based foods and lower in animal-based foods [96]. Red meat can also be culturally symbolic, representing strength and masculinity [97]. Combined, these factors may amplify resistance among frequent meat-eaters to purchase less meat.

Participants who had completed more than a high school degree had higher PME ratings than participants who had completed high school or less. Few studies have explored the effect of education on willingness to reduce meat purchases or consumption. Our findings are consistent with a nationally representative Dutch study that found that as education level increased, participants were more willing to eat meatless meals [97]. This finding complements other research that has found that having greater awareness of the environmental toll of meat increases willingness to eat less meat [62]. It may be that participants in our study with a higher education level were more discouraged from buying beef due to their greater knowledge of meat’s environmental harms. However, we did not assess this knowledge of harms in our study. This relationship warrants further exploration.

Despite sizable research that has indicated that being female is a predictor of willingness to eat less meat [62], we observed no relationship between gender and PME in our study. Income level also had no effect on participants’ PME ratings.

Strengths of this study include that it was the first study to our knowledge to experimentally test perceived effectiveness of different environmental messages aimed at reducing red meat consumption and that it involved a large sample of U.S. adults. Additionally, the sample was approximately half Latino, which allowed a large enough sample to explore the association between Latino ethnicity and responsiveness to environmentally themed messages to reduce meat purchases.

Limitations of this study include that the messages were displayed and assessed in an online survey, and as such consumption and behavioral outcomes were not assessed. Moreover, the design, while allowing us to compare two types of goal frames, precluded the ability to compare messages to a control. We also did not evaluate participants’ engagement level in other pro-environmental behaviors or emotional responses to the messages, both of which have been shown to influence which goal frame is more impactful at promoting pro-environmental behaviors [48,49] and could have helped explain why both goal frames performed similarly in this study. Finally, we did not assess participants’ baseline views on or knowledge of meat consumption and its effect on the environment, which would have helped to better contextualize the PME ratings overall. Future research should examine these types of messages in more detail (e.g., using qualitative methods) and test how they affect consumers’ purchases of red meat in real-world settings.

## 5. Conclusions

Reducing beef consumption in the U.S. is critical for promoting human and planetary health. This study is one of the first that has explored the impact of goal or emphasis framing on messages aimed at reducing red meat consumption. We found that messages emphasizing that purchasing less beef *reduces* environmental harms performed similarly to messages emphasizing that purchasing beef *worsens* environmental harms. There were few differences between specific environmental topics (e.g., water shortages and greenhouse gas emissions), though the message focusing on ‘the environment’ at large elicited the greatest response from survey participants overall and seems promising for future messaging studies. To build on this finding, additional research should assess whether the term ‘environment’ is more familiar or understandable to consumers than more specific environmental harms. We also recommend directly testing the effectiveness of emphasis frames spanning across multiple topics (e.g., health and environment) to determine which frame overall is best at discouraging red meat consumption.

Populations that were over 39 years old, ate beef more than once a week, had a high school education or less, and were non-Latino were less responsive to the messages overall. Collectively, these findings suggest that to maximize the effectiveness of messages to reduce red meat consumption, employing different messaging tactics for different demographics may be a winning strategy. To assess this, future research should test the effectiveness of these messages across other subpopulations and explore how these messages affect consumers’ purchases of red meat in real-world settings.

## Figures and Tables

**Figure 1 ijerph-19-02919-f001:**
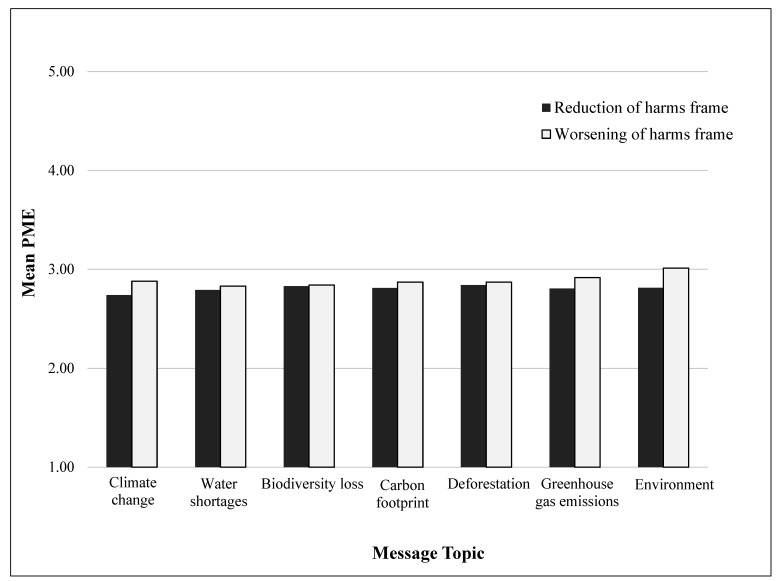
Perceived Message Effectiveness by Message Topic and Framing Condition in an Online Study of U.S. Adults (*n* = 1078).

**Figure 2 ijerph-19-02919-f002:**
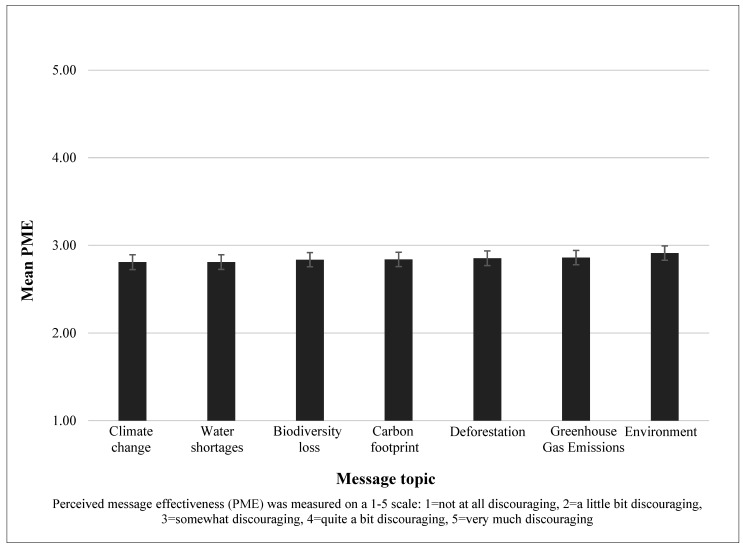
Perceived Message Effectiveness by Message Topic in an Online Study of U.S. Adults (*n* = 1078).

**Table 1 ijerph-19-02919-t001:** Text of Environmental Messages Tested in an Online Study of U.S. Adults (*n* = 1078).

Frame 1: Worsening of Environmental Harms	Frame 2: Reduction of Environmental Harms
Buying beef can…	Buying less beef can…
increase your carbon footprint	reduce your carbon footprint
increase greenhouse gas emissions	reduce greenhouse gas emissions
contribute to water shortages	reduce water shortages
hurt the environment	help the environment
worsen climate change	help mitigate climate change
contribute to biodiversity loss	reduce biodiversity loss
contribute to deforestation	reduce deforestation

**Table 2 ijerph-19-02919-t002:** Descriptive Statistics of Sociodemographic Characteristics, Beef Consumption, and Language Preference of Participants in an Online Study of U.S. Adults (*n* = 1078).

*Age ** (Years), Mean and SD	*n* or Mean	% or SD
	35.3	7.4
Education level ^†^, *n* and %		
High school diploma ^‡^ or less	512	47.5
4 year college degree or more	566	52.5
Income level, *n* and %		
Less than $25,000	213	19.8
$25,000 to $49,999	288	26.7
$50,000 to $74,999	202	18.7
$75,000 to $99,999	157	14.6
$100,000 or more	218	20.2
Latino ethnicity, *n* and %		
Non-Latino	564	52.3
Latino	514	47.7
Gender, *n* and %		
Male	445	41.3
Female	628	58.3
Transgender	5	0.5
Beef consumption ^†^, *n* and %		
Less than once a week	377	34.9
Once a week or more	701	65.0
Survey language, *n* and %		
English	924	85.7
Spanish	154	14.3

* Three incorrect age entries were treated as missing values. ^†^ Demographic data for these categories were initially collected using the following categories but were dichotomized to avoid small cells by treatment status and increase power by reducing the number of categories. For education, these categories were: less than high school or U.S. equivalent, high school or U.S. equivalent, 4 year college degree, and graduate degree or more. For beef consumption, these categories were: never, once a month, 2–3 times a month, once a week, 2–6 times a week, once a day, and more than once a day. ^‡^ Or U.S. equivalent (GED).

**Table 3 ijerph-19-02919-t003:** Results of Ordinary Least Squares Regression of Perceived Message Effectiveness on Demographic Covariates in an Online Study of U.S. Adults (*n* = 1078).

	Coefficient	*p* Value	95% CI	Standardized Coefficient
Aged over 39 (vs. between 18 and 29)	−0.28 *	0.013	−0.49, −0.06	−0.10
Aged between 30 and 39 (vs. between 18 and 29)	−0.06	0.507	−0.25, 0.12	−0.03
Eats beef more than once a week (vs. once a week or less)	−0.26 ***	0.001	−0.41, −0.11	−0.10
More than high school (vs. high school or less)	0.24 **	0.005	0.08, 0.41	0.10
Latino ethnicity (vs. non Latino ethnicity)	0.21 **	0.005	0.07, 0.36	0.09
Male (vs. female) †	0.06	0.455	−0.10, 0.21	0.02
Income between $35,000 and $74,999 (vs. less than $35,000)	−0.02	0.842	−0.20, 0.17	−0.01
Income of $75,000 or more (vs. less than $35,000)	0.19	0.068	−0.01, 0.40	0.07

* *p* < 0.05, ** *p* < 0.01, *** *p* < 0.001. † We do not report the coefficient for transgender (relative to female, the reference gender category) due to small cell size (*n* = 5).

## Data Availability

Data will be shared upon request; those requesting access will be added to the UNC IRB before receiving data.

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
