# Peer review of "Designing Environmental Messages to Discourage Red Meat Consumption: An Online Experiment"

_ijerph, 2022, doi:10.3390/ijerph19052919_

Round 1

Reviewer 1 Report

Thank you for addressing my concerns.

Author Response

Thank you for this comment. We appreciated your feedback!

Reviewer 2 Report

The authors have improved significantly the article, but they have to extend a little bit the Conclusions (they are too much short and it offers a not equlibrated final result). It is the question to optimize.

Author Response

Thank you for this comment and for your feedback. Details of how we addressed your feedback are included in the cover letter for our newly resubmitted manuscript.

Reviewer 3 Report

The authors' revised version reply to the reviewers' comments.  If other reviewers do not have further questions, I recommend to accept after minor revision. Well done.

Author Response

Thank you for this comment. We appreciated your feedback!

This manuscript is a resubmission of an earlier submission. The following is a list of the peer review reports and author responses from that submission.

Round 1

Reviewer 1 Report

This paper addresses a current issue, environmental messages. Also, the topic is original and innovative, as there is lack of background in the related sustainability literature. In our opinion, the content of this work is interesting and completed, although one of the biggest weaknesses we see is its abstract structure and literature review.

Abstract

The abstract shows a lot of number (n=? , mean=?....).  This is not academic style for the abstract section. ( please revise this for the following :

“....at discouraging red meat consumption 18 in the US. Adults (n=1078) viewed seven messages in an online survey highlighting the reduction 19 or worsening of environmental harms associated with eating red meat (between-subjects factor) and 20 rated the messages on how much they discouraged them from wanting to buy beef. Each messaged 21 highlighted a different environmental harm: deforestation, climate change, water shortages, biodi- 22 versity loss, carbon footprint, greenhouse gas emissions, or environment (within-subjects factor). 23 No statistically significant difference (p=.277) was found between the messages conveying the re- 24 duction (mean=2.80) or worsening (mean=2.89) of environmental harms for most topics, though the 25 worsening of harms frame ‘environment’ message (mean=3.01) outperformed the reduction of 26 harms ‘environment’ message (mean=2.80, .....”

1.Introduction

The subject of the study and its research objectives are presented properly. However, the author(s) should improve the work more synthesized. Furthermore, it should incorporate a description of the procedures used and the sequence of work for theoretical development. Contrary to what is stated in this section, this work does not present a true synthesis (or should be your intention) of the literature on the value generated for the mitigating climate change and preventing chronic disease. The reason is because few Sustainability literature cited including clear references  for the Sustainability subject.

2.Materials and Methods:

All major relevant terms are discussed at work.  The provision of paragraphs can be improved, and the structure of this Materials and Methods section. Moreover, the research equation and hypotheses are not clearly exhibited.

3.Results

The method selected for the empirical analysis is suitable for the study area. It is explained in a clear and understandable way. However, it should explain more specifically how the process of categorization of the variables used in the results of Ordinary Least Squares Regression is performed.

The article discusses the existence of several existing conditions in the model.  I suggest that the author(s)  should be appropriate to identify, explain and discuss its significance implications for the study.

References

The number of references is acceptable and major works that are related to the issues cited. However, the author(s) are missing some important publications on sustainability Journal such as  the topic of sustainability, meat consumption, environmental behavior and health communication in recent years.

Reviewer 2 Report

The proposal is attractive. However, to improve, you need the next questions:

-Theoretical framework: to update some references.

-Methods. The principal weak is the use of unique quantitative tools. I suggest to complete Methodology with a qualitative tool (Delphi, in this case, to analyse the results to avoid possible deviations). It will confer more entrust to the research.

-Results: it will be improved with the methodological new qualitative tools.

-Conclusion: too much brief and poor. You have to go in deep and contrast with the authors of the theoretical framework.

Reviewer 3 Report

Very interesting. About the only real criticism is that I would have liked a little more depth on the previous research, even given that there has not been a great deal done to this point. A more solid foundation rather than running almost straight to the experiment would be beneficial.